# Obeticholic Acid—A Pharmacological and Clinical Review

**Caezaan Keshvani [1], Jonathan Kopel [1,*] and Hemant Goyal [2]**

[1] Department of Medicine, Texas Tech University Health Sciences Center, Lubbock, TX 79430, USA
[2] Instructor in Medicine, Division of Gastroenterology, Hepatology and Nutrition, The University of Texas Health Science Center at Houston, Houston, TX 77030, USA
* Correspondence: jonathan.kopel@ttuhsc.edu

**Abstract:** Obeticholic acid (OCA) or 6-alpha-ethyl-chenodeoxycholic acid is a semisynthetic modified bile acid derivative that acts on the farnesoid X receptor (FXR) as an agonist with a higher potency than bile acid. The FXR is a nuclear receptor highly expressed in the liver and small intestine and regulates bile acid, cholesterol, glucose metabolism, inflammation, and apoptosis. The FXR group of bile acid receptors is currently under investigation for their potential role in the treatment of primary biliary cirrhosis (PBC), non-alcoholic steatohepatitis (NASH), and primary sclerosing cholangitis (PSC). Recent clinical studies suggest OCA may work synergistically with lipid modifying medications to further improve long-term outcomes with primary sclerosing cholangitis. Specifically, OCA can improve clinical outcomes in NASH patients with their different histological, metabolic, and biochemical issues as well as improve morbidity and mortality in patients suffering from PBC, PSC, or liver disease. This improvement is noted in both improved histological examination and reduced need for transplantation. In this review, we examine the pharmacology of OCA towards the treatment of PBC refractory and steatohepatitis (NASH). In addition, we examine future directions and applications of OCA for PBC, PSC, NASH, and NAFLD.

**Keywords:** obeticholic acid; non-alcoholic fatty liver disease; metabolic syndrome, biliary cirrhosis; primary biliary cholangitis; non-alcoholic steatohepatitis; primary sclerosing cholangitis

## 1. Introduction

The farnesoid X receptor (FXR) is a nuclear receptor that can form a heterodimer with the retinoid X receptor (RXR) or bind to its gene response element in its monomeric form. The binding to the FXR element (FXRE) can result in differential activation or repression of downstream targets [1–6]. The FXR is a nuclear receptor highly expressed in the liver and small intestine, especially the distal ileum, as well as the kidneys, adrenal glands, muscles, adipose tissue, and cardiac muscle, that regulates bile acid, cholesterol, glucose metabolism, inflammation, and apoptosis [1–4,6,7]. The role of FXR in liver and small intestine has been extensively studied in regulating bile acid, cholesterol, glucose metabolism, inflammation, and apoptosis [8]. In the liver, FXR primarily acts as a bile acid sensor. FXR activation causes release of FGF19, acts as an endocrine signaling molecule from enterocytes to the liver, which causes a decrease in bile acid synthesis and release. LRH1 and SHP levels are increased due to FXR activation, which also decreases bile acid synthesis via inhibition of CYP7A1, cholesterol utilization and fatty acid metabolism. Several animal and clinical studies are underway to target therapies, such as Obeticholic acid (OCA), towards FXR for the treatment of cardiovascular disease, male infertility, kidney failure, obesity, and vascular disease [9–11]. Obeticholic acid (OCA) or 6-alpha-ethyl-chenodeoxycholic acid is a semisynthetic modified bile acid (BA) derived from chenodeoxycholic acid that is a potent activator of the farnesoid X receptor (FXR) [1–4,12]. Figure 1 shows the effects of OCA on physiology and pathology where it crosses the cell membrane in liver and enterocytes to activate FXR that can form a heterodimer with the RXR, homodimer with FXR, or bind to DNA as a monomer.The FXR is a nuclear receptor highly

expressed in the liver and small intestine that regulates bile acid, cholesterol, glucose metabolism, inflammation, and apoptosis. Several animal and clinical studies are underway for targeted therapies towards FXR for the treatment of cardiovascular disease, male infertility, kidney failure, obesity, and vascular disease [9–11]. In the following sections, we will examine the effects of OCA on different physiological processes in the body, particularly within the liver.

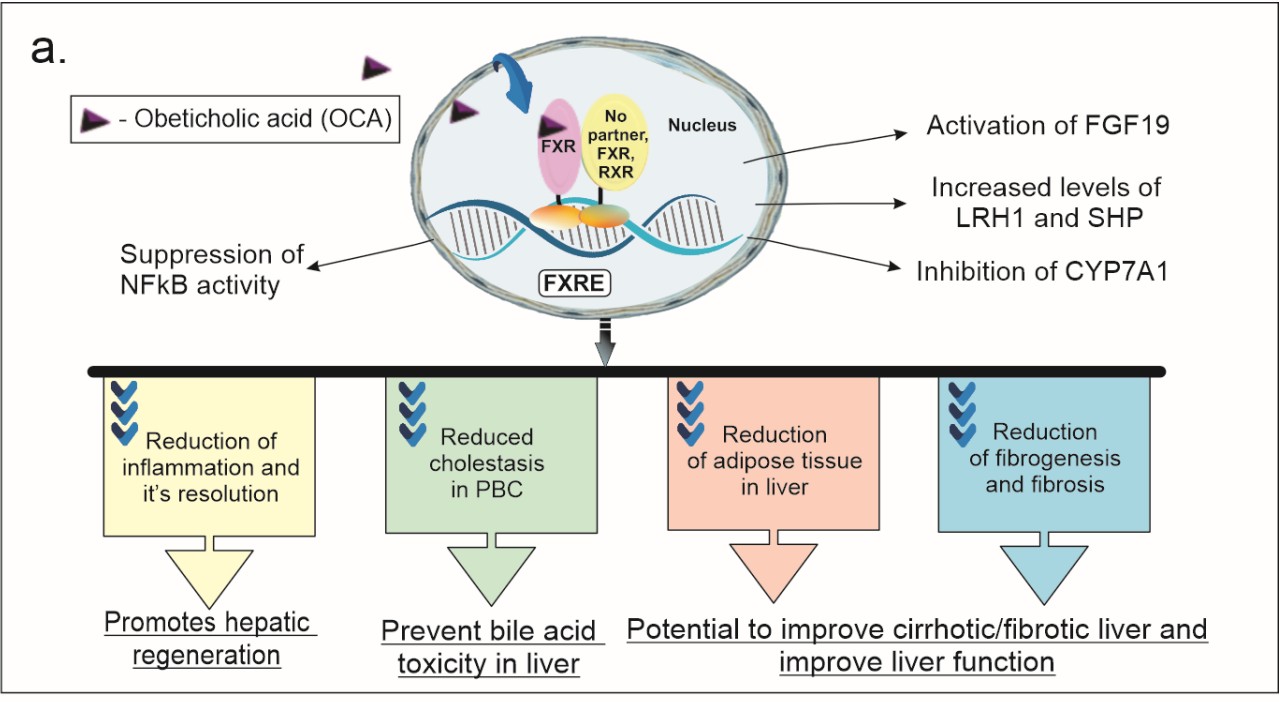

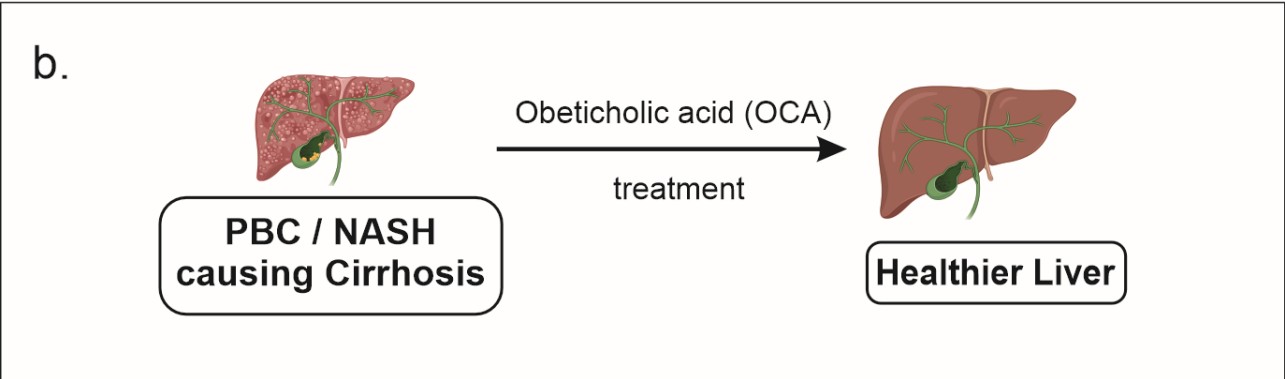

**Figure 1.** OCA effects on physiology and pathology: (**a**) OCA crosses the cell membrane in liver and enterocytes to activate FXR (FXR) that can form a heterodimer with the retinoid x receptor (RXR), a homodimer with FXR, or bind to DNA as a monomer. This can activate various signaling pathways, decreasing bile acid synthesis, fatty acid and cholesterol metabolism, glucose metabolism, inflammation and fibrosis. FXR activation causes release of fibroblast growth factor 19 (FGF19) which acts as an endocrine signaling molecule released from enterocytes to the liver causing decreased bile acid. Liver related homolog 1 (LRH1) and small heterodimer protein (SHP) levels are increased due to FXR activation, which also decreased bile acid synthesis via inhibition of CYP7A1, cholesterol utilization and fatty acid metabolism. This all improves cholestasis in patients with PBC. In mouse models, anti-inflammatory effects of OCA are mediated via suppression of nuclear factor k activator of B cells (NFkB) signaling. This mediates reduction in hepatic inflammation and fibrosis seen in mouse models of non-alcoholic steato hepatitis (NASH). (**b**) Improvement in liver functional status after OCA treatment in mouse models of cirrhosis due to NASH or primary biliary cholangitis (PBC).

## 2. The Effects of OCA on Different Physiological Processes through FXR Activation

### 2.1. OCA Effect on Bile Acid Synthesis

One of FXR's main roles is to inhibit bile acid synthesis through regulating the expression of small heterodimer partner (SHP). SHP inactivates liver-related homolog-1 protein (LRH-1), which in turn represses the expression of the cholesterol 7$\alpha$-hydroxylase (CYP7A1) enzyme [13]. CYP7A1 is the rate-limiting step in BA synthesis within the liver. Another mechanism by which FXR can downregulate CYP7A1 is via the induction of fibroblast growth factor 19 (FGF19) within enterocytes located in the ileal region. This molecule can act as an endocrine signaling hormone, activating the JNK signaling cascade, and causing a downregulation of CYP7A1 expression after reaching the liver via enterohepatic circulation [14,15]. Through inhibiting and downregulating CYP7A1, FXR is believed to exert hepatoprotective properties by preventing the accumulation of toxic BA buildup in the liver and promoting hepatic regeneration after  liver damage in mouse models [16–18].

The primary method by which OCA reduces circulating bile salts is through upregulation of the bile salt exporter pump (BSEP) to increase bile salt excretion into the biliary tree [19–21]. In addition, OCA decreases intestinal cholesterol absorption rather than increasing biliary cholesterol production [22]. OCA also inhibits hepatic sterol 12 hydroxylase (CYP8B1) and cholesterol 7 hydroxylase (CYP7A1), in part by inducing a small heterodimer partner [22]. This results in a smaller bile acid pool and a change in the composition of the bile, with an increase in the proportion of $\alpha/\beta$-muricholic acid and a decrease in taurocholic acid [22]. Additional studies using primary hepatocytes showed that OCA treatment suppressed bile acid synthesis genes (CYP7A1, CYP27A1) and increased bile efflux genes (ABCB4, ABCB11, OSTA, OSTB) [23,24].

### 2.2. OCA Effect on Fatty Acid Metabolism

In addition, the FXR plays an important role in fat metabolism, which includes regulating the expression of several lipoproteins, including apolipoprotein E and very low-density lipoprotein receptor (VLDLR) [25–30]. By phosphorylating insulin responsive substrate-1 on serine 312 in the liver and muscles, FXR activation was found to reduce body weight gain, prevent fat from accumulating in the liver and muscles, and reverse insulin resistance [26]. Genes involved in fatty acid production, lipogenesis, and gluconeogenesis had their liver expression decreased by FXR activation. Free fatty acid production in the muscles was also decreased by FXR therapy [26]. A subsequent study using OCA reduced the amount of visceral adipose tissue, steatosis, and other inflammatory markers in rabbits fed a high fat diet [31]. Similar results were also observed in mice, hamsters, and human [32–35].

### 2.3. OCA's Effect on Vascular and Inflammatory Processes in the Liver

Within vascular smooth muscle cells, OCA can induce smooth muscle cell death and downregulate interleukin (IL)-1$\beta$-induced inducible nitric oxide synthase and cyclooxygenase-2 expression [36]. In addition, OCA suppressed smooth muscle cell migration stimulated by platelet-derived growth factor-BB. However, a recent mouse model study suggested that OCA may not help with hepatic regeneration resulting from liver injury [37]. Further studies will need to be done to assess whether OCA may revert liver damage in human patients. The binding of OCA to FXR has also been shown to decrease the expression of nuclear factor $\kappa$ light-chain enhancer of activated B cells (NF- $\kappa$B) by inducing expression of cytochrome P450. This process can reduce inflammasome formation and prevent the development of various liver and vascular pathologies [38].

Along with its effects on metabolism, OCA also acts as an anti-inflammatory agent via activation of FXR in liver [36,39–44]. Several animal studies have shown that loss of FXR or FXR deficiency, results in increased hepatic inflammation and fibrosis [39–42,45,46]. Specifically, it was found that activation of FXR reduced the production of

fibrosis from hepatic stellate cells through altering the activities of peroxisome prolifera-tor-activated receptor gamma (PPARγ) and SHP. In response, there is a corresponding reduction in the pro-fibrotic activities of α1 collagen, TGF-β1, and NLR family pyrin do-main containing 3 (NLRP3) inflammasome activation [43,46]. Further improvements in hepatic inflammation and fibrosis were also observed when OCA was combined with glu-cagon-like peptide-1 receptor (GLP-1R) agonists [47]. A similar result was also observed when OCA was combined with an anti-apoptosis inhibitor (IDN-6556), PPAR-α/δ ago-nists, and statins [48–50]. Further studies using primary hepatocytes treated with OCA reduced TGFβ and IL-6 signaling pathways as well as reduced HDL genes (SCARB1, ApoAI, LCAT) while LDL genes (ApoB, CYP7A1) increased [23].

In recent years, the FXR group of bile acid receptors is currently under investigation for its potential role in the treatment of primary biliary colangitis (PBC), non-alcoholic steatohepatitis (NASH), non-alcoholic fatty liver disease (NAFLD), and primary scle-rosing cholangitis (PSC) [51–71]. In 2016, the Food and Drug Administration approved OCA for treating PBC refractory to ursodeoxycholic acid (UDCA) [45]. Besides OCA, sev-eral other FXR agonists are under investigation for the treatment of PBC and NASH [72–75]. Recent clinical studies suggest OCA may work synergistically with lipid modifying medications to further improve long-term outcomes with primary sclerosing cholangitis. Specifically, OCA improves morbidity and mortality in NASH patients with their differ-ent histological, metabolic, and biochemical issues in PBC, PSC, or liver patients. In this review, we examine the pharmacology of OCA towards the treatment of PBC, PSC, NASH, and NAFLD. In addition, we examine future directions and applications of OCA for PBC, PSC, NASH, and NAFLD.

### 3. Clinical Trial Studies of OCA in PBC, PSC, and NASH/NAFLD

*3.1. Primary Biliary Cholangitis*

Primary biliary cholangitis is characterized by the destruction of small intrahepatic bile ducts due to chronic inflammatory processes [76,77]. The interplay of genetic predis-position and environmental stimuli is believed to contribute to the development of PBC. The condition is very common among first-degree relatives, which suggests a genetic pre-disposition. Environmental factors can cause an autoimmune response in people who are genetically predisposed, which activates both the humoral and cellular responses against ducts [76,77]. Small interlobular bile duct cells are destroyed as a result of B cells and T cells fighting antigens that invade the liver and begin attacking bile ductular cells. Hepato-cytes are destroyed when bile duct cells are destroyed because this prevents bile from draining from canaliculi, which causes cholestasis. Following this cholestatic hepatitis, cirrhosis and progressive fibrosis develop [76,77]. Serum alkaline phosphatase (ALP) is the most potent prognostic marker in PBC, but in diagnoses initially abnormal liver serum tests are noticed [78,79]. Since 1997, UDCA has been the only drug that the FDA approved to treat PBC. However, many PBC patients are either refractory or lose response to UDCA treatment [80]. OCA's mechanism of action is distinct from UDCA and has been shown to improve ALP levels when combined with UDCA [81]. Since its discovery, several clinical trials have examined the use of OCA for primary biliary cirrhosis (Table 1).

A clinical trial by Nevens et al., known as the POISE trial, tested whether OCA re-duced ALP and bilirubin levels. The POISE trial found that daily dosages of up to 50 mg of OCA led to greater decreases in ALP and bilirubin levels [59]. However, there was a corresponding increase in dose-related pruritus, especially at dosages greater than 10 mg. Considering these results, Neven's et al. examined the longer-term effectiveness, safety, and adverse events of 5-10 mg OCA in patients with PBC [51]. In this study, patients par-ticipated in a 12-month double-blind, placebo-controlled study [51]. Patients were given a once-daily oral placebo, or OCA at a dose of 5 mg that was increased to 10 mg, or OCA at a dose of 10 mg [51]. Overall, the POISE trial found that, at 12 months, there was no

significant difference in liver fibrosis between either the treatment or the placebo group [51]. In contrast, PBC patients treated with OCA reported a higher incidence of pruritus.

A post hoc analysis using POISE trial data showed improvement in more than one liver disease stage in 37% of the 5-10 mg OCA and 35% of the 10 mg OCA patients, respectively, compared to the placebo group (12%) [82]. Furthermore, disease progression occurred in 10% and 0% of the 5–10 mg and 10 mg OCA groups, respectively, compared to 37% of patients in the placebo group [82]. Taken together, the post-hoc analysis of the POISE trial supported OCA's efficacy in reducing the risk of liver-related complications in both PSC and PBC patients [82]. A separate analysis was conducted to examine whether there was a difference in the first occurrence of liver transplantation or death in PBC patients treated with OCA in the POISE trial compared to comparable non-OCA-treated external controls from the Global PBC Registry (Global PBC) and United Kingdom PBC Registry (UK-PBC) [70]. Even with the external controls in the Global PBC and UK-PBC registry, PBC patients who received OCA had improved overall survival and transplant-free survival compared to the control group.

**Table 1.** Primary Biliary Cholangitis OCA Clinical Trials.

| Author / Trial Name | Study Type | Number of Patients | Outcomes |
|---|---|---|---|
| Nevens et al. [51] (POISE trial) | RCT * | 217 | • Dose-dependent pruritus observed with OCA<br>• OCA decrease ALP and total bilirubin<br>• No change in PSC or PBC fibrosis<br>OCA had improved transplant-free survival<br>• Reduced mortality risk for liver disease patients<br>• OCA therapy reduces the risk of liver-related complications in PSC and PBC patients<br>• Interim analysis showed persistent treatment efficacy after three years using OCA |
| D'Amato et al. [52] | Retrospective | 191 | • OCA decreased ALP, alanine transaminase, and bilirubin<br>• Patients with cirrhosis treated with OCA showed lower efficacy in reducing liver fibrosis due to reduced tolerability and higher discontinuation rate<br>• Patients with a significant liver fibrosis who took OCA had worse pruritus<br>• OCA improved disease progression in one-third of patients who did not respond to ursodeoxycholic acid |
| Kowdley et al. [54] | RCT | 59 | • Reduced ALP, c-glutamyl transpeptidase, alanine aminotransferase, conjugated bilirubin, and immunoglobulin M<br>• OCA increased circulating FGF-19 and decreased C4 levels<br>• Dose-dependent pruritus observed with OCA<br>• At six years, OCA improved cholestasis, hepatocellular damage, and liver function |

* RCT – randomized control trial.

After completion of the POISE trial, a separate retrospective study analyzed data from 191 patients in the Italian PBC Registry to assess the real-world efficacy of OCA [52]. At 12 months, there was a decrease in ALP, alanine transaminase, and bilirubin. Using criteria in the POISE TRIAL, the response rate of PBC patients using OCA was 42.9%; in contrast, criteria examining measurements in liver function tests and gallbladder function (e.g. alanine aminotransferase, ALP, and bilirubin) showed that the response rate for PBC patients using OCA was only 11% [52]. Patients with cirrhosis showed a lower reduction in liver fibrosis, which was believed to be due to reduced tolerability and a higher discontinuation rate of patients using OCA [52]. However, most patients with PBC still required additional therapy to normalize liver biochemistry even with OCA therapy [52]. PBC patients also reported increased pruritus, especially in patients with a past medical history of cirrhosis [52]. Overall, OCA was found to be helpful in more than one-third of patients

who did not respond to ursodeoxycholic acid [52]. A subsequent three-year interim analysis of the POISE trial showed that ALP concentrations were reduced compared with baseline at 12, 24, 36, and 48 months [63]. In addition, total bilirubin concentrations were decreased at 12 and 48 months [63]. Direct bilirubin also decreased at 12 months [63]. However, total and direct bilirubin did not change. OCA was generally well tolerated despite complaints of puritus. No serious adverse events were related to OCA [63]. The three-year interim analyses showed long-term efficacy and safety of OCA in patients with PBC who were intolerant or unresponsive to ursodeoxycholic acid [63].

Kowdley et al. also examined the safety, tolerability, and efficacy of daily OCA in 59 patients using 10 or 50 mg OCA [54]. The study evaluated the efficacy, safety, and durability of treatment with OCA in an ongoing open-label extension with up to 6 years of treatment [54]. The study found that ALP was reduced in both OCA groups [54]. Similar reductions in liver function tests were observed through 6 years of open-label extension treatment [54]. OCA treatment also improved markers of cholestasis, hepatocellular damage, and liver function, which persisted through 6 years of open-label treatment [54]. Patients treated with OCA also showed increased circulating FGF-19 and decreased C4 levels, which confirmed FXR activation in patients treated with OCA. Similar to the POISE trial, pruritus was the most common adverse side effect in the 10 mg and 50 mg OCA groups. The symptoms disappear after discontinuation of OCA [54]. Overall, cholestasis, hepatocellular damage, and liver function improved using OCA even after six years of extended therapy [54].

### 3.2. Primary Sclerosing Cholangitis

Primary sclerosing cholangitis is a cholestatic liver illness that is chronic and progressive and has no known cause [83]. PSC is characterized by inflammation, fibrosis, and/or stricturing of intra- and/or extrahepatic biliary ducts. PSC often develops over time leading to cholestasis and liver failure. Although the cause of primary sclerosing cholangitis is unknown, environmental and genetic variables are believed to contribute to its pathogenesis [83]. In addition, inflammatory bowel disease (IBD) has been shown to increase the risk of a patient developing PSC. As such, PSC is thought to be an autoimmune condition. Some individuals have high levels of antineutrophilic cytoplasmic antibodies, antinuclear antibodies, and anticardiolipin antibodies [83]. Additionally, the condition is more likely to affect carriers of the HLA-B8 and HLA-DR3 genes. Inflammation, fibrosis, and cholestasis are the hallmarks of primary sclerosing cholangitis [83].

A randomized clinical trial, known as the AESOP trial (Assessment of Efficacy and Safety of OCA in PSC), by Kowdley et al. used a randomized, double-blind, placebo-controlled trial to assess the efficacy and safety of OCA in PSC patients (Table 2) [53]. After completing the initial stage of the trial, patients were placed in an open-label, long-term safety extension [53]. In the trial, a total of 76 patients were randomly assigned 1:1:1 treatment to three groups: 1.5 mg titrating to 3.0 mg OCA, 5 mg titrating to 10 mg OCA, or placebo [53]. After which, patients reconsented to the long-term safety extension, with visits scheduled every 3 months for up to an additional 24 months [53]. All patients, including those who had received placebo during, were given OCA treatment at 5 mg, except for those who completed treatment with 10 mg OCA [53]. At 24 weeks, patients that were administered OCA at 5–10 mg showed a decrease in serum ALP compared to placebo [53]. In contrast, patients given OCA at 1.5–3.0 mg did not significantly lower serum ALP at 24 weeks [53]. Similar to other clinical studies, dose-related pruritus was the most frequent adverse reaction to OCA therapy [53]. During the long-term safety extension, reductions in ALP achieved with OCA therapy were maintained [53]. Overall, the AESOP trial showed that OCA reduced serum ALP during an initial 24-week treatment period, which was maintained for a 2-year, long-term extension of the study.

**Table 2.** Primary Sclerosing Cholangitis OCA Clinical Trials.

| Author / Trial Name | Study Type | Number of Patients | Outcomes |
|---|---|---|---|
| Kowdley et al. [53] (AESOP trial) | RCT | 76 | • Decreased ALP at 5 – 10 mg; no change in ALP at 1.5-3.0 mg<br>• Dose-dependent pruritus observed with OCA<br>• Reductions in ALP persisted for 2 years using OCA |

*3.3. Non-Alcoholic Steatohepatitis and Non-Alcoholic Fatty Liver Disease*

In addition to PBC and PSC, OCA has shown effectiveness in treating both non-alcoholic fatty liver disease (NAFLD) and non-alcoholic steatohepatitis (NASH) (Table 3). Non-alcoholic fatty liver disease (NAFLD) refers to a variety of conditions that are characterized by the presence of hepatic steatosis on imaging or histology (macro-vesicular steatosis) and the absence of secondary causes of hepatic steatosis [84]. The majority of the time, NAFLD is discovered by accident during imaging or when it presents with symptoms, such as pruritus. The onset and progression of non-alcoholic fatty liver disease (NAFLD) are influenced by both environmental and genetic variables. Patients with NAFLD who have first-degree relatives are more at risk than the general public. cAMP-responsive element-binding protein H (CREBH) or sirtuin regulates gene expression by maintaining the chromatin structure and amino-terminal ends of histones (SIRT1). According to genetic research, SIRT1 activation may contribute to the emergence of NAFLD. Through aberrant DNA methylation, NAFLD is initiated, which leads to cancer [84].

In 1998, Day and James suggested a two-hit pathogenesis model. The initial blow is brought on by insulin resistance, which causes triglyceride droplets to build up in the cytoplasm of hepatocytes, resulting in steatosis [85,86]. Due to decreased elimination and increased transport of free fatty acids and triglycerides to the liver, insulin resistance results in buildup. Additionally, an abundance of carbs stimulates the liver's production of de novo fatty acids. Hepatocellular damage brought on by the second strike and the emergence of NASH are complex. The liver is more susceptible to damage when there are too many fatty acids present [85,86]. Hepatocellular damage brought on by the second strike and the emergence of NASH are complex. The liver is more susceptible to damage when there are too many fatty acids present. The injury is thought to be caused by peroxisomal fatty acid oxidation, reactive oxygen species (ROS) production from the mitochondrial respiratory chain, cytochrome P450 fatty acid metabolism, and hepatic metabolism of gut-derived alcohol. In the second hit, insulin resistance is also included. NASH develops and progresses as a result of sinusoidal collagen deposition brought on by the activation of hepatic stellate cells and portal fibrosis brought on by ductular proliferation. These alterations have been linked to insulin resistance, which is now thought to be the driving factor behind the development of progressive fibrosis and NASH from steatosis. In response, hepatocytes experience cytoskeletal aggregation, ballooning, apoptosis, and necrosis [85,86].

Currently, there is no therapy available for treatment of NASH [87]. Several clinical studies are underway for NASH treatment, including OCA. One of the mechanisms of action of OCA has been to reduce hepatic inflammation and steatosis has been shown in multiple mouse models [16]. Currently, there have been three trials that have examined the use of OCA for NASH and NAFLD [55–58].

**Table 3.** Non-alcoholic fatty liver disease or non-alcoholic steatohepatitis OCA clinical trials.

| Author / Trial Name | Study Type | Number of Patients | Outcomes |
|---|---|---|---|
| Younossi et al. [55,56] (REGENERATE trial) | RCT | 913 | • Patients receiving 25 mg of OCA showed improvements in hepatocellular ballooning and lobular inflammation<br>• Dose-dependent pruritus observed with OCA |
| Neuschwander-Tetri et al. [57] (FLINT trial) | RCT | 283 | • OCA improved fibrosis, hepatocellular ballooning, steatosis, and lobular inflammation on histological examination<br>• After 36 weeks, OCA decreased blood alanine aminotransferase and aspartate aminotransferase concentrations; in contrast, gamma-glutamyl transpeptidase levels increased while serum ALP levels decrease<br>• OCA decreased with weight loss and systolic blood pressure<br>• OCA therapy initially increased total serum cholesterol and LDL cholesterol, and decreased HDL cholesterol; cholesterol levels decreased and returned to normal after discontinuing OCA<br>• OCA did not impact overall VLDL particle concentration; however, it reduced big VLDL particle concentration<br>• Genetic markers at chromosomes 1, 4, 6, 7, 15, and 17 influenced treatment efficacy and side effects |
| Pockros et al. [58] (CONTROL trial) | RCT | 84 | • 5 mg, 10 mg or 25 mg OCA increased low-density lipoprotein cholesterol (LDLc) and mean LDL particle concentration (LDLpc)<br>• 10 mg atorvastatin with OCA decreased LDLc and LDLpc levels |

Younossi et al. examined OCA treatment in patients with NASH and NAFLD in a clinical trial known as the randomized global phase 3 study to evaluate the impact on NASH with fibrosis of OCA treatment (REGENERATE trial), [55,56]. The REGENERATE trial is a multicenter, randomized, double-blind, placebo controlled trial and was conducted at 332 centers in 20 countries, and included a total of 1,968 patients [55,56]. Patients were randomized to receive daily placebo, 10 mg OCA, or 25 mg OCA [55,56]. After 18 months, the study found that 12% of the placebo group's patients, 18% of the OCA 10 mg group, and 23% of the 25 mg OCA group, showed improvement in liver fibrosis [55,56]. In addition, patients receiving 25 mg OCA showed improvements in hepatocellular ballooning and lobular inflammation [55,56]. Among the patients receiving OCA, pruritus was the most frequent side effect experienced, which improved over time or after discontinuing OCA [55,56]. Overall, the REGENERATE trial found that treatment with OCA using a 25 mg dose improved fibrosis and prevented progression of fibrotic disease [55,56].

A clinical trial by Neuschwander-Tetri et al., known as the Farnesoid X nuclear receptor ligand OCA for non-cirrhotic, non-alcoholic steatohepatitis (FLINT trial), also examined the efficacy of OCA in patients with NASH [57]. The FLINT trial consisted of a multicenter, randomized trial with a total duration of 72 weeks of treatment using either OCA or placebo in patients with NASH [57]. Among the 283 patients in the study, OCA was randomly given to 141 patients, while a placebo was given to 142 [57]. As opposed to 21% in the placebo group, 45% of patients in the OCA group showed improved liver histology [57]. In addition, more patients who were given OCA showed improvement in fibrosis, hepatocellular ballooning, steatosis, and lobular inflammation [57]. Patients receiving OCA also showed a greater average change in the NAFLD activity score than those receiving a placebo [57]. Despite these improvements in the specific histological characteristics of NASH, there was no difference between the proportion of patients whose NASH resolved between those receiving OCA or the placebo [57]. Over the first 36 weeks of OCA administration, there was a persistent decrease in blood alanine aminotransferase, aspartate aminotransferase, and ALP levels in the blood in patients receiving OCA [57].

In contrast, gamma-glutamyl transpeptidase levels increased [57]. After the OCA was discontinued, the alanine aminotransferase and aspartate aminotransferase concentrations returned to baseline; after 24 weeks discontinuing therapy, there were no discernible differences between either group [57]. Subsequent analysis of patients receiving OCA showed OCA improved weight loss and decreased systolic blood pressure [57]. Furthermore, patients who received treatment with OCA had higher concentrations of total serum cholesterol and LDL cholesterol, and a decrease in HDL cholesterol [57]. These changes developed within 12 weeks of beginning treatment, which diminished over time [57]. Similar to other clinical trials using OCA, 23% of patients receiving OCA had pruritus compared to 6% in the placebo group [57].

A post hoc analysis of the 283 patients in the FLINT trial confirmed that OCA improved weight reduction, serum aminotransferases, and histology [62]. Specifically, patients who received OCA experienced larger changes in their fibrosis stage. The highest changes were seen in patients who also lost weight, indicating that the benefits of the OCA treatment and weight loss were additive [62]. Another study found that FLINT trial patients on OCA did not impact overall VLDL particle concentrations; however, it was linked to a reduction in big VLDL particle concentration after 12 weeks when compared to the placebo [64]. After 12 weeks, the OCA group had a greater overall LDL particle concentration than the placebo group [64]. Large and medium HDL particle concentrations also decreased in the OCA treated individuals as compared to placebo [64]. However, these alterations in LDL, VLDL, and HDL disappeared after OCA was discontinued. A genome wide analysis of the FLINT trial patient cohort showed multiple single nucleotide polymorphisms (SNPs) in six locations on chromosomes 1, 4, 6, 7, 15, and 17 were correlated with NASH resolution among patients who received OCA [87]. Of these SNPs, those near the TDRD10 and ANO3 genes on chromosomes 1 and 11, respectively, were associated with improved portal inflammation and NASH among patients receiving OCA therapy [87]. In addition, the rs75508464 variant near CELA3B had the most significant effect on NASH resolution in those receiving OCA.

Lastly, a recent clinical trial, known as The Combination OCA aNd sTatins for monitoRing Of Lipids (CONTROL trial), was a randomized, double-blind, placebo-controlled trial designed to assess the effect of statins with OCA therapy on the lipid profile of NASH patients [58]. The study design included a screening period of up to 5 weeks (including a 4-week statin washout period), a 16-week double-blind treatment phase, and an open-label long-term safety extension up to 2 years [58]. A total of 84 patients were randomly assigned in a 1:1:1:1 to receive the placebo, 5 mg, 10 mg or 25 mg OCA once daily during the 16-week double-blind phase [58]. After 4 weeks, all OCA groups had an increase from baseline in mean low-density lipoprotein cholesterol (LDLc) and mean LDL particle concentration (LDLpc), mostly owing to large, less atherogenic LDLc particles [58]. Furthermore, 10 mg atorvastatin decreased LDLc and LDLpc levels below baseline in all OCA groups by 8 weeks. However, increased doses did not provide additional clinical benefits [58]. Overall, the CONTROL trial showed that OCA increases in LDL levels among NASH patients were reduced using atorvastatin. The combination of OCA and atorvastatin was generally safe and well tolerated.

## 4. Discussion

FXR is a novel target for regulating several biochemical processes involved in bile acid and cholesterol synthesis, glucose metabolism, inflammation, and apoptosis. FXR is expressed throughout the body and primarily serves as a bile acid sensor in the liver. FGF19, an endocrine signaling molecule that is secreted from enterocytes to the liver and results in a decrease in bile acid, is increased by FXR activation. OCA is a well-tolerated drug that does not disrupt the activity, metabolism, or excretion of other medications. By upregulating the bile salt excretion in the biliary system through activating FXR, OCA reduces bile acid synthesis through a variety of methods, improving outcomes in PBC patients. FXR activation slows the metabolism of fatty acids and cholesterol, which may

help NASH. Additionally, FXR activation reduces fibrosis and suppresses inflammation, which benefits the development and progression of PBC and NASH.

By activating FXR, OCA is a novel treatment that can be used in conjunction with other drugs to improve clinical outcomes in for PBC, PSC, NASH, and NAFLD. These processes are crucial for understanding the pathogenesis and treatment of PBC, PSC, NASH, and NAFLD. Clinical studies in NASH patients have developed to use NASH and fibrosis improvement as key outcomes. As such, the development of advanced disease in NASH can be improved by reducing hepatic fibrosis. OCA is a novel and very promising medication that can help NASH patients with their different histological, metabolic, and biochemical issues. In addition, the ability of OCA to lower ALP and bilirubin levels leads to improved morbidity and mortality in patients suffering from biliary or liver disease. This improvement reduces the need for liver transplantation. In addition, specific genetic markers may improve the activity of OCA or reduce the onset of pruritus. For most patients, OCA is well-tolerated.

## 5. Conclusions

Recent clinical studies suggest that OCA may work synergistically with lipid modifying medications to further improve long-term outcomes with primary sclerosing cholangitis. Furthermore, OCA is suitable for patients who have persistent elevations in ALP that are resistant to ursodeoxycholic acid therapy. Further clinical trials are needed to assess which combinations of medications improve clinical outcomes for PBC, PSC, NASH, and NAFLD. Given its effectiveness in these pathologies, future clinical trials examining the effectiveness of OCA for the treatment of cardiovascular disease, male infertility, kidney failure, obesity, and vascular disease are warranted.

**Author Contributions:** Conceptualization, J.K. and H.G. writing—original draft preparation, J.K., C.K. and H.G.; writing—review and editing, J.K., C.K. and H.G., supervision, H.G. All authors have read and agreed to the published version of the manuscript.

**Funding:** This research received no external funding.

**Institutional Review Board Statement:** Not applicable.

**Informed Consent Statement:** Not applicable.

**Data Availability Statement:** Not applicable.

**Conflicts of Interest:** The authors declare no conflict of interest.

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
