# Peer review of "Obeticholic Acid—A Pharmacological and Clinical Review"

_futurepharmacol, doi:10.3390/futurepharmacol3010017_

Round 1

Reviewer 1 Report

The manuscript submited by Keshvani et al. is a narrative review over the promising results yielded in human trials in terms of the use of OCA as treatment for NASH and PBC and PSC, however, many suggested changes must be taken into acount.

line 54 a space is missing in afterreading...

line 75 flora should be replaced by microbiota

line 77  the word that should be replaced by which 

line 82 the word release should be included after decreased

In figure 1, it is not clear what is the role of OCA during phisiology as authors stated. Also, in the yellow inset the sentence is confusing. In pink inset, the authors mean fat surrounding the liver or steatosis? Blue inset, what is the difference between fibrogenesis and fibrosis

line 101 responCe is misspelled

line 102 authors used ALP  but line 97 alkaline phosphatase. the rest of the manuscript is inconsistent with other acronyms as well OCA, UDCA lines126 and 128.  

lines 96 authors refer to PBC as primary biliary cholangitis (the correct one) and then as primary biliary cirrhosis line 104

line 132 was a difference is duplicated

line 107 higher decreases is confusing

and so on...

Author Response

The manuscript submited by Keshvani et al. is a narrative review over the promising results yielded in human trials in terms of the use of OCA as treatment for NASH and PBC and PSC, however, many suggested changes must be taken into acount.
-We appreciate the reviewer’s comments. We went through the entire manuscript and expanded and/or changed the wording or grammar where appropriate. We also addressed the specific grammar points addressed by this reviewer.
line 54 a space is missing in afterreading...
We appreciate the reviewer’s comments. We made the appropriate changes to the manuscript.
line 75 flora should be replaced by microbiota
We appreciate the reviewer’s comments. We made the appropriate changes to the manuscript.
line 77  the word that should be replaced by which
We appreciate the reviewer’s comments. We made the appropriate changes to the manuscript.
line 82 the word release should be included after decreased
We appreciate the reviewer’s comments. We made the appropriate changes to the manuscript.

In figure 1, it is not clear what is the role of OCA during phiysiology as authors stated.

Thank you for the question. The role of OCA is important for treating patients that have NASH and/ or PBC. We are trying to show the mechanism of action of OCA in these patients since it activates the FXR which has multiple transcriptional targets. Some of these targets are mentioned in this review that are relevant for the effects observed in RCT and animal models. The effects of these targets include suppression of NFkB activity and activation of SHP- important for decreasing the inflammation response, activation of SHP causes inhibition of CYP7A1 which decreases BA production and thus improving steatosis,

FXR stimulates the synthesis of fibroblast growth factor-19 (FGF-19), which in turn participates in theinhibition of CYP7A1 and CYP8B1 expression through the fibroblast growth factor receptor- 4 (FGFR4) pathway in hepatocytes. As a result, the above-described FXR/SHP and FXR/FGF19/FGFR4 pathways are major negative regulators of BA synthesis. Furthermore, FXR inhibits the sodium taurocholate co-transporting polypeptide (NTCP) via SHP, thereby repressing hepatic BA uptake (Floreani et al ,2022)

Also, in the yellow inset the sentence is confusing.

The inset has been clarified since the purpose was to highlight reduction in inflammation.

In pink inset, the authors mean fat surrounding the liver or steatosis?

We meant steatosis since a number of studies in animal models have shown regression of steatosis when treated with OCA

Blue inset, what is the difference between fibrogenesis and fibrosis

We had mentioned fibrogenesis since NASH as time progresses can result in chronic inflammatory damage and stimulation of cells to lay down extracellular matrix protein which will result in fibrosis. Fibrosis is an end stage that can occur in these patients and studies have shown improvement in fibrosis with OCA administration.

line 101 responCe is misspelled
We appreciate the reviewer’s comments. We made the appropriate changes to the manuscript.
line 102 authors used ALP  but line 97 alkaline phosphatase. the rest of the manuscript is inconsistent with other acronyms as well OCA, UDCA lines126 and 128. 
We appreciate the reviewer’s comments. We made the appropriate changes to the manuscript.
lines 96 authors refer to PBC as primary biliary cholangitis (the correct one) and then as primary biliary cirrhosis line 104
We appreciate the reviewer’s comments. We made the appropriate changes to the manuscript.
line 132 was a difference is duplicated
We appreciate the reviewer’s comments. We made the appropriate changes to the manuscript.
line 107 higher after decreases is confusing
We appreciate the reviewer’s comments. We made the appropriate changes to the manuscript.

Reviewer 2 Report

This review looked at the pharmacology of obeticholic acid in the treatment of PBC refractory and steatohepatitis.

The paper is overall well-written and the content is interesting to the readers. I suggest accepting it in its present form. 

Author Response

Comments and Suggestions for Authors
This review looked at the pharmacology of obeticholic acid in the treatment of PBC refractory and steatohepatitis.The paper is overall well-written and the content is interesting to the readers. I suggest accepting it in its present form.

We appreciate the reviewer’s comment and taking time to review the manuscript.

Reviewer 3 Report

The authors have conducted a brief review of obeticholic acid and its roles in the treatment of different liver diseases. The topic is interesting but the manuscript is lacking "story" and "flow" in general. The information collected is very impressive but not presented in the best way. The manuscript must be significantly revised to improve readability and focus before it can be published. Below are some of the suggestions that the authors may want to consider for their next revision.

General:

The abstract is written in an interesting way and with proper flow. A similar structure and flow must be followed for the remaining manuscript, particularly in the conclusion. For example, since the topic is on OCA, the abstract starts with OCA, then MOA, then diseases applications, whereas in the conclusion it goes from diseases to MOA to diseases to OCA which is confusing. Moreover, the authors then decided to throw in contraindications that were never discussed all over the manuscript. The conclusions must be simply a brief overview of the manuscript. 

Any manuscript must be written in the following order.

Introduction - introduce what you are going to say and why you want to say it or why it is important

Discussion - say what you want to say

Conclusion - summarize what you said

The manuscript is lacking this format of writing.

Additionally, there are several typos and grammatical errors/sentence formation issues and professional words of choice that must be fixed all over the manuscript.

Below are some of the specific comments:

Abstract: 

Ln 9 - abbreviate as OCA and keep using OCA

Ln 10 Please abbreviate Farnesoid x receptor at first use as FXR.

Ln 15 define FXR first

Ln 16 - define PBC

NASH is Nonalcoholic steatohepatitis 

Introduction:

Ln 41 - BA receptor? abbreviates for what? bile acid?

Ln 54 - typo

Ln 55 - typo

Ln 59-72 there are multiple ideas presenting FXR roles that do not lead to any conclusion, not sure what point the authors are trying to make here

Ln 76-86 is redundant to MOA of OCA and figure 1 should be cited in the first paragraph of MOA. 

Figure 1 is confusing as to the figures presented at the bottom for NASH and PBC/NASH. The figure makes it look like OCA promotes these two while the opposite is true, the authors may want to show two different pictures of the liver - one with NASH and the other with NASH resolution and OCA in between.

Ln 105 - why is the POISE trial bolded? no need to bold - please do the same for all the other trials reported

Ln 203-204 can the authors add this pathway to figure 1? how does OCA mechanistically reduces inflammation? The authors may use a different figure to show the OCA MOA for inflammation

For tables, please left justify them to improve the readability

Ln 203-209 are not well connected to each other, looks like a list. This paragraph is lacking flow.

Conclusion

Ln 286-287 sentence is not clear. What does NAS stand for? NAFLD Activity Score?

Ln 292 OCA is medication?

Ln 297 should be the end of the conclusion. Not sure why there is another paragraph of discussion. This is not well written, is out of focus, and has different ideas. Either delete it or move it to the discussion above in the section where it fits. Or add a new section called "adverse effects".

Will look into references in the revised version. 

The authors have done impressive work on an interesting topic and I am looking forward to the revised version.

Author Response

Comments and Suggestions for Authors
The authors have conducted a brief review of obeticholic acid and its roles in the treatment of different liver diseases. The topic is interesting but the manuscript is lacking "story" and "flow" in general. The information collected is very impressive but not presented in the best way. The manuscript must be significantly revised to improve readability and focus before it can be published. Below are some of the suggestions that the authors may want to consider for their next revision.

-We appreciate the reviewer’s comment. The introduction and other portions of the manuscript were improved to help with the flow and story.

The abstract is written in an interesting way and with proper flow. A similar structure and flow must be followed for the remaining manuscript, particularly in the conclusion. For example, since the topic is on OCA, the abstract starts with OCA, then MOA, then diseases applications, whereas in the conclusion it goes from diseases to MOA to diseases to OCA which is confusing.

- We appreciate the reviewer’s comment. We made sure that the conclusion followed the same organization and focus as what was written in the abstract.

Moreover, the authors then decided to throw in contraindications that were never discussed all over the manuscript.

-We appreciate the reviewer’s comment. We removed the contraindications from the manuscript.

The conclusions must be simply a brief overview of the manuscript.

-We appreciate the reviewer’s comment. We made sure to focus the conclusion on summarizing the major portions and conclusions from the manuscript.

Any manuscript must be written in the following order.
Introduction - introduce what you are going to say and why you want to say it or why it is important
Discussion - say what you want to say
Conclusion - summarize what you said
The manuscript is lacking this format of writing.

-We appreciate the reviewer’s comment. The introduction is listed in sections 1 and 2. Section 3 is the discussion portion of the manuscript. Section 4 is the conclusion of the manuscript. With the extra information and reorganization of the manuscript, we believe that the flow and organization is improved from the original edit.

Additionally, there are several typos and grammatical errors/sentence formation issues and professional words of choice that must be fixed all over the manuscript.

-We appreciate the reviewer’s comment. We went through the entire manuscript and improve the grammar, writing, and content.

Abstract:
Ln 9 - abbreviate as OCA and keep using OCA
Ln 10 Please abbreviate Farnesoid x receptor at first use as FXR.
Ln 15 define FXR first
Ln 16 - define PBC
NASH is Nonalcoholic steatohepatitis

-We appreciate the reviewer’s comment. The abstract was completely re-written to help with the overall content to review.

Introduction:
Ln 41 - BA receptor? abbreviates for what? bile acid?
Ln 54 - typo
Ln 55 - typo
Ln 59-72 there are multiple ideas presenting FXR roles that do not lead to any conclusion, not sure what point the authors are trying to make here
Ln 76-86 is redundant to MOA of OCA and figure 1 should be cited in the first paragraph of MOA.

-We appreciate the reviewer’s comment. The abstract was completely re-written to help with the overall content to review.

Figure 1 is confusing as to the figures presented at the bottom for NASH and PBC/NASH. The figure makes it look like OCA promotes these two while the opposite is true, the authors may want to show two different pictures of the liver - one with NASH and the other with NASH resolution and OCA in between.

Thank you for the observation. We have modified the figure to incorporate your suggestion and made it so that it is clear about the potential of OCA to treat patients with NASH /PBC and improving their liver function.

Ln 105 - why is the POISE trial bolded? no need to bold - please do the same for all the other trials reported

-We appreciate the reviewer’s comment. All the trial names were un-bolded as requested.

Ln 203-204 can the authors add this pathway to figure 1? how does OCA mechanistically reduces inflammation? The authors may use a different figure to show the OCA MOA for inflammation
Thank you for the comment and observation. It has been shown that OCA via FXR activation targets a number of downstream genes responsible for controlling a number of different physiological processes from cholesterol metabolism to inflammation. An example of this is activation of SHP which has been shown in Zou et al (2018) in reducing inflammation in mouse model of NASH.  Our focus in this review was not solely on inflammation so we were summarizing some of the important functions of OCA. We can make the references clearer in our text that contain animal models and primary research.

(Zou, A., Magee, N., Deng, F., Lehn, S., Zhong, C., Zhang, Y., 2018. Hepatocyte nuclear receptor SHP suppresses inflammation and fibrosis in a mouse model of nonalcoholic steatohepatitis. Journal of Biological Chemistry 293, 8656–8671.. doi:10.1074/jbc.ra117.001653)

For tables, please left justify them to improve the readability
We left justified the table. The journal may be best to help with this since the overall format and look was changed after submission.

Ln 203-209 are not well connected to each other, looks like a list. This paragraph is lacking flow.

-We appreciate the reviewer’s comment. We completely re-wrote lines 203-209 in the manuscript as requested.

Conclusion
Ln 286-287 sentence is not clear. What does NAS stand for? NAFLD Activity Score?
Ln 292 OCA is medication?

Ln 297 should be the end of the conclusion. Not sure why there is another paragraph of discussion. This is not well written, is out of focus, and has different ideas. Either delete it or move it to the discussion above in the section where it fits. Or add a new section called "adverse effects".

-We appreciate the reviewer’s comment. The conclusion was rewritten and expanded. The term NAS was removed along with NAFLD score. In addition, the conclusion was focused on the additional applications of OCA for the treatment of other conditions mentioned in the introduction of the manuscript. The mention on adverse event was also removed as well.

Will look into references in the revised version.

-We appreciate the reviewer’s comment. We went through the literature again on the manuscript and significantly added references to make sure that all the relevant literature in the basic science and clinical trial portions was included.

The authors have done impressive work on an interesting topic and I am looking forward to the revised version.

--We appreciate the reviewer’s comment.  We hope that our changes and revisions help with the overall framing of the manuscript.

Round 2

Reviewer 1 Report

The manuscript has substantially been improved.

Author Response

Thank you. We appreciate the reviewer's comments. They helped to improve the manuscript. 

Reviewer 3 Report

The authors have revised the manuscript but have not made all the changes they mentioned in the response to the review.

Figure 1 appears to be the same - The only modification was the word "treatment" added. Not at all helpful and does not resolve the issue mentioned in the earlier review. In fact, now the figure is not even readable (poor graphics).

The authors may need to revisit my previous comments and try to implement each comment carefully. If the authors do not wish to accept/make changes, please provide a scientific justification with appropriate references in the response to review.

The tables are not justified even though the authors claim that they made these changes. 

The conclusion still includes side effects - a claim that the authors made that they were removed. The conclusion is still too long and provides another set of discussion points.

Not sure if the correct version of the manuscript was uploaded

The authors are responsible for checking every point-by-point response and making sure that those changes were implemented. Such issues raise concern over the accuracy and credibility of the authors.

Author Response

This contains our responses to the first and second round of revisions from reviewer one. We went through all the comments again from the beginning

Reviewer 1 – Original

The authors have conducted a brief review of obeticholic acid and its roles in the treatment of different liver diseases. The topic is interesting but the manuscript is lacking "story" and "flow" in general. The information collected is very impressive but not presented in the best way. The manuscript must be significantly revised to improve readability and focus before it can be published. Below are some of the suggestions that the authors may want to consider for their next revision.

We appreciate the reviewer’s comment. We expanded the paper on the introduction to improve the data provided for OCA and the FXR receptor. We expanded the role of FXR in bile acid synthesis. In addition, we provided additional citations on the preclinical studies on OCA’s mechanism with regards to reducing bile acid synthesis, lipid synthesis, inflammatory, and liver fibrosis. After which, we provided additional information for each section on OCA’s use with regards to the pathogenesis of the disease and the results of the clinical trials.

The abstract is written in an interesting way and with proper flow. A similar structure and flow must be followed for the remaining manuscript, particularly in the conclusion. For example, since the topic is on OCA, the abstract starts with OCA, then MOA, then diseases applications, whereas in the conclusion it goes from diseases to MOA to diseases to OCA which is confusing. Moreover, the authors then decided to throw in contraindications that were never discussed all over the manuscript. The conclusions must be simply a brief overview of the manuscript. 

-We appreciate the reviewer’s comment. We rewrote the abstract to follow

Any manuscript must be written in the following order.

Introduction - introduce what you are going to say and why you want to say it or why it is important

Discussion - say what you want to say

Conclusion - summarize what you said

The manuscript is lacking this format of writing.

We appreciate the reviewer’s comment. We added a section on the discussion as asked. We believe this helped to shorten the conclusion and organize the paper as requested.  

Additionally, there are several typos and grammatical errors/sentence formation issues and professional words of choice that must be fixed all over the manuscript.

-We appreciate the reviewer’s comments. The grammar of the manuscript was improved throughout the manuscript and double checked by the authors to make sure that the proper content and formatting was used throughout the paper.

Abstract: 

Ln 9 - abbreviate as OCA and keep using OCA

-We appreciate the reviewer’s comment. We expanded the word OCA on line 9. We made sure to remove obeticholic acid from the rest of the manuscript and replaced it with OCA.

Ln 10 Please abbreviate Farnesoid x receptor at first use as FXR.

-We appreciate the reviewer’s comment. We changed the word “Farnesoid X Receptors” to FXR at Lines 54 and 73

Ln 15 define FXR first; Ln 16 - define PBC; NASH is Nonalcoholic steatohepatitis 

-We appreciate the reviewer’s comments. Both PBC and FXR were defined on lines 15 and 16 respectively. We also corrected NASH in the manuscript as well.

Introduction:

Ln 41 - BA receptor? abbreviates for what? bile acid?

We appreciate the reviewer’s comments. We expanded the word BA on line 41 to make sure it says “bile acid”

Ln 54 - typo

We appreciate the reviewer’s comments. We fixed the typo in the previous edit.

Ln 55 – typo

We appreciate the reviewer’s comments. We fixed the typo in the previous edit.

Ln 59-72 there are multiple ideas presenting FXR roles that do not lead to any conclusion, not sure what point the authors are trying to make here

-We appreciate the reviewer’s comment. We reorganized the introduction to focus on FXR and then added sentences to transition into OCA as a novel agonist to target FXR for several liver pathologies.

Ln 76-86 is redundant to MOA of OCA and figure 1 should be cited in the first paragraph of MOA. 

We appreciate the reviewer’s comment. We removed this section from the previous edit

Figure 1 is confusing as to the figures presented at the bottom for NASH and PBC/NASH. The figure makes it look like OCA promotes these two while the opposite is true, the authors may want to show two different pictures of the liver - one with NASH and the other with NASH resolution and OCA in between.

We appreciate the reviewer’s comment. We separated Figure 1 into two separate figures. Figure 1 emphasizes the mechanism of action of OCA through FXR. Figure 1b shows the effect on OCA on the pathogenesis of liver disease.

Ln 105 - why is the POISE trial bolded? no need to bold - please do the same for all the other trials reported

-We appreciate the reviewer’s comment. Each mention of a trial’s name was unbolded in the manuscript.

Ln 203-204 can the authors add this pathway to figure 1? how does OCA mechanistically reduces inflammation? The authors may use a different figure to show the OCA MOA for inflammation

-We appreciate the reviewer’s comment. We decided to improve the clarity of the figure by splitting it into two subfigures to focus on the mechanism of action of OCA and its effects on the liver. We also made sure to show that the inflammation reduced by OCA is done through suppression of nuclear factor k activator of B cells (NFkB) signaling. This is listed in the figure legend.

For tables, please left justify them to improve the readability

-We appreciate the reviewer’s comment. We changed the table dimension sizes as well. In addition, it should be right justified; not left justified.

Ln 203-209 are not well connected to each other, looks like a list. This paragraph is lacking flow.

-We removed lines 203-209 and replaced them with two-three paragraphs on the pathophysiology of NASH/NAFLD.

Conclusion

Ln 286-287 sentence is not clear. What does NAS stand for? NAFLD Activity Score?

We appreciation the reviewer’s comment. This was removed from the last edit.

Ln 292 OCA is medication?

We appreciation the reviewer’s comment. This was removed from the last edit.

Ln 297 should be the end of the conclusion. Not sure why there is another paragraph of discussion. This is not well written, is out of focus, and has different ideas. Either delete it or move it to the discussion above in the section where it fits. Or add a new section called "adverse effects".

We appreciate the reviewer’s comment. We reorganized the conclusion into a separate discussion section to help with the organization of the manuscript in summarizing the main ideas and conclusions. We also removed any reference to “adverse effects”

Will look into references in the revised version. 

We appreciate the reviewer’s comment. We added around 30-40 references from the last edit, mainly in the mechanism of OCA and its effects on different metabolic and physiological processes within the cell

Reviewer 1 New Comments

The authors have revised the manuscript but have not made all the changes they mentioned in the response to the review. Figure 1 appears to be the same - The only modification was the word "treatment" added. Not at all helpful and does not resolve the issue mentioned in the earlier review. In fact, now the figure is not even readable (poor graphics).

-We agree with this assessment of the figure. We changed Figure 1 to improve the readability of the figure. We also added an extensive figure legend to help describe the figure content. We also split the figure into two portions to emphasize the mechanism of OCA in Figure 1A and its effects on the liver in Figure 1B.

The authors may need to revisit my previous comments and try to implement each comment carefully. If the authors do not wish to accept/make changes, please provide a scientific justification with appropriate references in the response to review.

-We appreciate the reviewer’s comments. We made sure to go through the manuscript again and respond to your original comments to ensure that it was properly done. We made significant additions, which altered the line numbers from our original edit. However, we made sure to provide justifications if we disagreed with any of the comments.

The tables are not justified even though the authors claim that they made these changes. 

-We appreciate the reviewer’s comment. We did try to justify the tables. However, the tables were large enough that doing so did not alter them significantly. In response, we modified the dimensions to make it easier to see the table’s justified in this edit.

The conclusion still includes side effects - a claim that the authors made that they were removed. The conclusion is still too long and provides another set of discussion points.

We appreciate the reviewer’s comments. We removed any mention of side effects in the conclusion. We also separated the conclusion into a discussion and short conclusion section as requested.

Not sure if the correct version of the manuscript was uploaded

-We uploaded the corrected verions to the manuscript. We mad significant changes to the areas added to each section (e.g. pathophysiology for PBC, NASH, NAFLD, ect). The large amount of red coloring could have made this difficult to see. We will call this manuscript version 3 to avoid any confusion.

The authors are responsible for checking every point-by-point response and making sure that those changes were implemented. Such issues raise concern over the accuracy and credibility of the authors.

-We appreciate the reviewer’s comment. We made sure to include the previous comments in this revision to ensure that each point was addressed to avoid any confusion.